# Screening and Management of Dyslipidemia in Children and Adolescents

**DOI:** 10.3390/jcm11216479

**Published:** 2022-10-31

**Authors:** Juliette M. Schefelker, Amy L. Peterson

**Affiliations:** Department of Pediatrics, Division of Pediatric Cardiology, School of Medicine and Public Health, University of Wisconsin, Madison, WI 53792, USA

**Keywords:** dyslipidemia, pediatric, atherosclerotic cardiovascular disease, familial hypercholesterolemia, cholesterol screening, universal screening

## Abstract

This review provides an overview of pediatric dyslipidemia emphasizing screening and treatment recommendations. The presence of risk factors for cardiovascular disease in childhood poses significant risk for the development of atherosclerotic cardiovascular disease and cardiovascular events in adulthood. While atherogenic dyslipidemia is the most common dyslipidemia seen in children and can be suspected based on the presence of risk factors (such as obesity), familial hypercholesterolemia can be found in children with no risk factors. As such, universal cholesterol screening is recommended to identify children with these disorders in order to initiate treatment and reduce the risk of future cardiovascular disease. Treatment of pediatric dyslipidemia begins with lifestyle modifications, but primary genetic dyslipidemias may require medications such as statins. As pediatric lipid disorders often have genetic or familial components, it is important that all physicians are aware that cardiovascular risk begins in childhood, and can both identify these disorders in pediatric patients and counsel their adult patients with dyslipidemia to have their children screened.

## 1. Introduction

Atherosclerotic cardiovascular disease (ASCVD) and its long-term sequelae are the leading cause of death worldwide [1]. The pathologic process of atherosclerosis begins in youth and increases risk for cardiac events such as heart disease, myocardial infarction, and stroke later in life. Risk factors for ASCVD that are found in childhood include lifestyle factors, medical conditions which increase risk (such as obesity or diabetes), as well as genetic conditions which increase lipid levels, such as Familial Hypercholesterolemia (FH). The purpose of this article is to discuss screening and treatment recommendations for pediatric dyslipidemias. It is known that ASCVD risk factors present during childhood are very likely to track into adulthood and are associated with increased risk for cardiovascular events in adulthood [2,3]. Recently, this association between childhood risk factors and adult ASCVD events was demonstrated [4]. Children with an average age of 11.8 years underwent evaluation including body mass index (BMI), total cholesterol (TC), triglycerides (TG), systolic blood pressure (SBP), and smoking and were reevaluated 35 years later. At a mean age of 46 years, 3.8% of participants had an ASCVD event, and 0.8% had a fatal event. The association between events and risk factors was significant for each individual risk factor, and was even greater when combining risk factors. Importantly, this study demonstrated duration of cholesterol elevation is predictive of ASCVD events. This highlights the importance of early recognition and intervention in pediatric patients to prevent future ASCVD events. 

As pediatric dyslipidemias can be identified via screening and have long-lasting impact, we present the following summary of the different disorders and evidence-based treatments to help facilitate care to patients and families. It is important to note that normal lipid and lipoprotein concentrations are different between children and adults. Pediatric reference ranges are listed in Table 1 [5].

## 2. Heterozygous Familial Hypercholesterolemia (HeFH)

Familial Hypercholesterolemia is an autosomal dominant genetic condition that results in elevated LDL-C levels starting at birth. In its heterozygous form, it is the most common severe genetic dyslipidemia in pediatric patients. A global meta-analysis estimated the prevalence in the general population is 1 in 311 [6], while data in the US have shown rates to be as high as 1 in 250 adults [7].

### 2.1. Diagnosis

Several sets of criteria have been developed to diagnose HeFH; however, most were developed primarily for adults and so their use in children can be limited (Table 2). All sets of criteria rely on elevated LDL-C and many incorporate family history and physical exam findings indicative of elevated cholesterol (primarily tendinous xanthomas or arcus cornealis). However, physical exam findings associated with hypercholesterolemia are vanishingly rare in children with HeFH. In fact, if such physical findings are identified in a child, the clinician should consider rarer dyslipidemias, including homozygous familial hypercholesterolemia, sitosterolemia, or cerebrotendinous xanthomatosis as more likely than HeFH.

For ease of diagnosis, the American Heart Association recommended clinical criteria for diagnosis of HeFH in children, including LDL-C ≥ 160 mg/dL in a child with family history of elevated cholesterol or premature ASCVD in a parent or grandparent, or LDL-C ≥ 190 mg/dL (irrespective of family history), once secondary causes of hypercholesterolemia are excluded [8]. Genetic testing can be utilized to aid in diagnosis but is not required to make a clinical diagnosis of HeFH. However, a recent study found an association between receiving a genetic diagnosis of FH and willingness to be treated with a statin medication, suggesting a genetic diagnosis of HeFH may be perceived differently by patients and their families [9].

**Table 2 jcm-11-06479-t002:** Diagnostic criteria for heterozygous familial hypercholesterolemia in children [10]. Reprinted with permission from Peterson AL, McNeal CJ, and Wilson DP. Prevention of atherosclerotic cardiovascular disease in children with familial hypercholesterolemia. Curr Atheroscler Rep. 2021 Aug 27;23(10):64. 2021, Springer Nature.

Simon Broome criteria [11]				
Definite or Probable diagnosis of HeFH requires elevated cholesterol: Total cholesterol > 260 mg/dL or LDL-C > 155 mg/dL if ≤ 15 years Total cholesterol > 290 mg/dL or LDL-C > 190 mg/dL if ≥ 16 years PLUS One or more additional findings: Definite HeFH: additional findings 1. Tendon xanthoma in the child, first-degree relative, or second-degree relative 2. Genetic testing of a confirmed pathogenic variant (LDLR, ApoB, or PCSK9) POSSIBLE HeFH: additional findings 1. Family history of myocardial infarction ≤ 60 years in a first-degree relative or ≤50 years in a second-degree relative 2. Family history of a total cholesterol ≥ 290 mg/dL in a first or				
MEDPED criteria [12]				
A child is considered to have HeFH if total cholesterol meets or exceeds the threshold listed below. Thresholds vary based upon whether or not there is a first-, second-, or third-degree relative known to have HeFH.				
Child’s age	Does the child have one or more relatives with HeFH?			
≤19 years	Yes			No
	First degree	Second degree	Third degree	N/A
Total cholesterol	≥220 mg/dL	≥230 mg/dL	≥240 mg/dL	≥270 mg/dL
Dutch lipid clinic network criteria [13]				
Diagnosis of HeFH is based on the total number of points obtained. Definite HeFH, >8 points. Probable HeFH, 6–8 points. Possible HeFH, 3–5 points. Unlikely HeFH, <3 points				
Criterion:				Points:
Family history:				
First-degree relative with known premature ASCVD (<55 years in men, <60 years in women), OR first-degree relative with LDL-C ≥ 95%ile				1
First-degree relative with tendinous xanthomata and/or arcus cornealis, OR pediatric first degree relative with LDL-C ≥ 95%ile				2
Clinical history:				
Patient with premature ASCVD (<55 years in men, <60 years in women)				2
Patient with premature cerebral or peripheral vascular disease				1
Physical Examination:				
Tendinous xantomata				6
Arcus cornealis with onset prior to 45 years				4
Patient’s cholesterol levels:				
LDL-C ≥ 330 mg/dL				8
LDL-C 250–329 mg/dL				5
LDL-C 190–249 mg/dL				3
LDL-C 155–189 mg/dL				1
Genetic testing				
Pathogenic variant in LDLR, APOB, or PCSK9				8
American Heart Association criteria [8]				
Children (≤18 years) with LDL-C ≥ 160 mg/dL AND Family history of elevated cholesterol or premature ASCVD AND No evidence of secondary causes of hypercholesterolemia				

### 2.2. Treatment of Pediatric HeFH

The first step in treatment of any pediatric dyslipidemia is addressing lifestyle factors, which may be exacerbating dyslipidemia, and accelerating risk factor development (Table 3). Specifically, for pediatric HeFH and for patients with other forms of elevated LDL-C, it is important to emphasize a diet that limits saturated fats, trans fats, and dietary cholesterol [5,14]. If these initial dietary modifications are unsuccessful, adherence to the CHILD-2 LDL-C diet [5] with further restriction of saturated fat and dietary cholesterol as well as emphasizing increased fiber intake may provide further LDL-C lowering benefit [14]. These measures should be implemented with the goal of optimizing diet and exercise for at least 3–6 months before considering medications in most circumstances. 

Although pharmacotherapy is rarely required to treat most pediatric dyslipidemia, HeFH is the most common indication for lipid lowering therapy use in pediatrics. Similar to adults, most pediatric patients with HeFH will require medications to meet LDL-C reduction goals. The first line agents for pediatric HeFH are statins, all of which are FDA-approved for use in pediatrics. 

The vast majority of pediatric HeFH patients achieve LDL-C reduction goals with statin monotherapy. Before a statin is considered, most eligible pediatric patients have LDL-C ≥ 160 mg/dL after 3–6 months of lifestyle modifications. For most pediatric patients with HeFH, statin dose is titrated to LDL-C level with the goal of LDL-C < 130 mg/dL on treatment [17]. Table 4 shows pediatric-specific considerations when prescribing statins. 

Statins have been found to be safe and effective in pediatric populations, with studies showing effective LDL-C reduction and minimal side effects in the short to intermediate term [13,17,18,19,20,21,22]. Luirink et al. (2019) performed a 20-year follow-up of pediatric HeFH patients treated with pravastatin and found that these patients had lower rates of ASCVD- related cardiac events than their affected parents who had not started statins until early adulthood. Of the 213 participants followed, only four discontinued the medication due to side effects, and no participants reported serious adverse effects such as rhabdomyolysis [22]. Anagnostis et al. (2020) found no adverse events related to statin use, with over 50% of patients meeting LDL-C reduction goals with a high-dose statin [20].

If statin monotherapy does not sufficiently lower LDL-C levels, other medications can be considered as secondary agents. The most common non-statin agent used for pediatric HeFH is ezetimibe, which is FDA approved for youth 10 years of age and older for HeFH. Its most common application is for additional LDL reduction while on statin therapy; however, it can be used as monotherapy for pediatric FH. Evolocumab, a proprotein subtilisin/kexin type 9 (PCSK9) inhibitor, is approved for additional LDL-C reduction in pediatric patients with HeFH who are 10 years and older. The HAUSER-RCT study demonstrated 44.5% reduction in LDL-C compared to placebo in pediatric patients aged 10–17 years old on a background of stable lipid lowering therapy, with similar incidence of adverse events between evolocumab and placebo [24,25].

Other medications used to treat adults with HeFH are currently undergoing pediatric trials. The ODDYSSEY KIDS trial of alirocumab in pediatric patients with HeFH demonstrated LDL-C reductions of 45% in individuals taking higher doses with favorable adverse effect profiles [26]. The ORION-16 trial investigating inclisiran vs. placebo for treatment of pediatric HeFH is underway [27].

## 3. Atherogenic Dyslipidemia

Atherogenic dyslipidemia is the most common dyslipidemia in childhood and is highly associated with childhood obesity or metabolic syndrome, affecting 33% of overweight and 43% of obese children [28]. Similar to adults, it is also commonly found in children who have insulin resistance, type 2 diabetes mellitus, and/or non-alcoholic fatty liver disease (NAFLD) [5,23,29,30,31]. 

### 3.1. Diagnosis

A fasting lipid panel can readily diagnose atherogenic dyslipidemia. The results are characterized by elevations in TG levels and decreased levels of HDL-C, a pattern similar to adults. In children, LDL-C levels are generally normal, although the LDL that is present is in the form of the more atherogenic small dense LDL particles. Importantly, the normal ranges for TG are different for children compared to adults; for children 0–9 years old, TG < 75 mg/dL is normal, and for children and adolescents 10–19 years old, TG < 90 mg/dL is normal [5]. See Table 1 for pediatric lipid and lipoprotein values.

### 3.2. Treatment of Pediatric Atherogenic Dyslipidemia

Treatment is primary through lifestyle changes, with a major focus on diet and activity modifications. For children with excess body fat, a modest decrease in weight has been shown to significantly reduce TG levels as well as increase HDL-C levels [32,33].

Sedentary lifestyle is a significant concern for children. The Physical Activity Guidelines for Americans recommend 60 min daily of moderate to vigorous physical activity along with muscle strengthening exercise 3 days per week for children aged 6–17 years old. Children aged 3–5 are encouraged to be physically active throughout the day via active play [15].

Dietary changes are a key consideration for improving cardiometabolic health. All dietary changes should be discussed through shared decision making with parents or guardians, as well as the child to be sure that cultural norms as well as cost and access to food are considered. Throughout the lifespan, focus on a variety of nutrient-dense foods and minimizing foods and beverages with excess added sugars is essential. When treating dyslipidemia, initial focus on adhering to diet suggested by the Dietary Guidelines for Americans can be helpful [16]. If there is insufficient improvement, it may be appropriate to consider a more restrictive diet. For children with elevated TG, limiting saturated fats and refined carbohydrates has been found to be effective [34]. With this in mind, minimizing intake of sugar-sweetened beverages as much as possible is fundamental when treating children, as these often are inadvertent but significant sources of sugar and empty calories. Increasing fiber and omega-3-fatty acid intake are helpful [14]. Working with a registered dietician can be extremely beneficial in ensuring that families are provided with comprehensive plans [35]. 

Very little evidence exists to guide the use of nutritional supplements in pediatric dyslipidemias, and the focus of treatment is generally a healthy diet. Further information regarding supplements can be found at the following resource [34].

## 4. Mixed Dyslipidemia

Pediatric patients with a mixed dyslipidemia have more modest elevations in LDL-C and TG, and generally normal HDL-C. They often have a clinical picture that is neither clearly attributable to lifestyle factors nor a known genetic diagnosis such as FH. Using the diagnosis of mixed dyslipidemia in these circumstances has the utility of helping to guide clinical decision making for these patients, as the dyslipidemia in these children often has an environmental component, but genetic etiologies should also be considered, particularly if patients do not respond to lifestyle therapy. 

### 4.1. Diagnosis

On a lipid panel, these patients typically have elevations in LDL-C, but not to the level which would be expected in HeFH, as well as elevations in TG. HDL-C is generally normal.

### 4.2. Treatment

As the clinical picture may appear unclear, treatment is a combination of previously described strategies. Lifestyle factors must first be optimized to help determine next steps in management. Re-assessment of lipid levels after initial changes can determine if the initial modifications were adequate or if further treatment is necessary. If LDL-C levels continue to be sufficiently elevated despite 3–6 months of interventions, initiation of pharmacotherapy with statins may be appropriate. Greater insight into when statin treatment is recommended based on a patient’s risk factors can be found at the following resource [23].

## 5. Rare Lipid Disorders

### 5.1. Homozygous Familial Hypercholesterolemia

Homozygous familial hypercholesterolemia (HoFH) can present in childhood and affects approximately 1 in 160,000 to 1 in 400,000 individuals. The disease often comes to clinical attention when children develop xanthomas, or it can be found through cholesterol screening. Children with HoFH will have LDL-C levels typically ranging from 500 mg/dL to 1,000 mg/dL and can develop overt coronary artery disease in the first decade of life.

Treatment of these individuals in childhood uses strategies similar to those used in adulthood and focuses on aggressive LDL-C reduction from the time of diagnosis. While HeFH can be managed in the primary care setting, pediatric lipid specialists should be consulted if HoFH is suspected. Treatment options often depend on whether or not affected individuals have functional LDL-receptor (LDL-R) activity. For those with functional LDL-R activity, statins and ezetimibe have been the primary therapy. Recently, evolocumab was approved by the FDA for patients 13 years of age or older with HoFH. Other therapies act independently from the LDL receptor. Lomitapide can be used to lower LDL-C but carries risk of hepatotoxicity. Evinacumab, an inhibitor of angiopoietin-like 3 protein (ANGTPL3) is approved for use in individuals 12 years and older for treatment of HoFH. LDL apheresis has traditionally been a mainstay of therapy but has additional challenges associated with pediatric use, including vascular access, circulating blood volume, and patient cooperation. Liver transplant has been used to treat this disease [36]. 

### 5.2. Severe Hypertriglyceridemia

As with adults, pediatric hypertriglyceridemia can be divided into primary and secondary causes. Most pediatric patients with hypertriglyceridemia have atherogenic dyslipidemia as described above. Important secondary causes of hypertriglyceridemia in pediatric patients include hypothyroidism, kidney disease (nephrotic syndrome), diabetes, liver disease, hypercortisolism, and medications. Pregnancy and excessive alcohol intake are important differentials in pediatric patients as well as adults. A comprehensive list of medications associated with hypertriglyceridemia is found in the 2018 AHA/ACC Multisociety Guideline on the Management of Blood Cholesterol [23]. Medications causing hypertriglyceridemia that are most commonly encountered in pediatrics include isotretinoin, L-asparaginase, oral estrogens, glucocorticoids, atypical antipsychotics, and immunosuppressive agents like tacrolimus, sirolimus, and cyclosporine. 

Primary hypertriglyceridemia in children is associated with severe elevations in fasting triglycerides, generally ≥500 mg/dL, although many have triglycerides ≥1000 mg/dL. Genetic testing should be considered to determine the underlying etiology of the disorder, as therapeutic lifestyle changes are the primary form of therapy but vary according to the underlying diagnosis. For familial chylomicronemia syndrome, caused by mutations in lipoprotein lipase, a specialized very low-fat diet is needed to prevent pancreatitis [37]. For individuals with other forms of hypertriglyceridemia, most commonly familial combined hyperlipidemia, dietary modifications are focused on reducing sugar and simple carbohydrate intake. In these cases, prescription omega-3 fatty acids (DHA and EPA in combination or EPA-only formulations) are used off-label to treat adolescents with hypertriglyceridemia, although they are not FDA-approved for this indication [38].

### 5.3. Hypobetalipoproteinemia and Abetalipoproteinemia

Very low LDL-C is occasionally diagnosed in pediatric patients. It is not generally associated with higher risk of ASCVD. Acquired causes of low LDL-C such as malignancy, malabsorption, medications, and severe illness or infection should be excluded. There is a group of very rare disorders that can cause very low levels of LDL-C, usually ≤25 mg/dL but many ≤10 mg/dL. They include homozygous hypobetalipoproteinemia (caused by mutations in apolipoprotein B) and abetalipoproteinemia (caused by mutations in microsomal triglyceride transfer protein). Typically, these children will present to medical attention with symptoms such as fat malabsorption, failure to thrive, hepatomegaly, and manifestations of fat-soluble vitamin deficiencies [39]. Management focuses on monitoring for and treating fat-soluble vitamin deficiencies and monitoring for hepatic steatosis [40].

Individuals who are heterozygous for hypobetalipoproteinemia have LDL-C levels below average and are usually asymptomatic. They are generally thought to be at reduced risk for ASCVD, but they could develop hepatic steatosis. No treatment is generally indicated but individuals should be monitored occasionally for steatohepatitis.

## 6. Lipoprotein(a)

Lipoprotein (a) [Lp(a)] is an LDL-C moiety with an ApoB protein covalently bound to apolipoprotein (a). Plasma levels of Lp(a) are incredibly variable and are 70- 90% determined by genotype. The prevalence of elevated Lp(a) in children is presumably the same as for adults, as there is thought to be little variability in Lp(a) levels throughout the lifespan. There are very few studies in children, but data indicate that high Lp(a) in a child is a risk factor for arterial ischemic stroke [41] and venous thromboembolism [42]. Pediatric values for Lp(a) are the same as those used for adults.

There is no FDA-approved therapy for treatment of elevated Lp(a) in children, and medications shown to lower Lp(a) in adults, namely PCSK9 inhibitors, are only used very rarely in children. Therefore, any efforts to screen children for elevated Lp(a) must balance concerns about privacy, autonomy, and provoking anxiety against the potential benefits of future therapies. The National Lipid Association identifies four pediatric groups in which Lp(a) testing is reasonable: (1) Clinically suspected or genetically confirmed familial hypercholesterolemia; (2) Family history of a first-degree relative with premature ASCVD (<55 years in men, <65 years in women); (3) Pediatric ischemic stroke with unknown cause; or (4) Parent or sibling with elevated Lp(a) [43].

## 7. Screening for Pediatric Dyslipidemias

Fundamentally, the purpose of lipid screening in childhood is to identify children with dyslipidemia in order to pursue early treatment through lifestyle modification and/or medical management and decrease risk of ASCVD events in adulthood. Several different screening strategies exist, including selective, universal, and cascade screening. 

### 7.1. Selective Screening

Selective screening involves screening children with high-risk medical conditions or with family history that increases their likelihood of developing ASCVD. High-risk medical conditions commonly encountered in children include obesity, diabetes, and elevated blood pressure, as well as less common conditions like childhood cancer, solid organ transplantation, Kawasaki disease with persistent aneurysms, kidney disease, and some forms of congenital heart disease. Selective screening should also be considered for a child with a significant family history of early cardiovascular disease or diabetes. Selective lipid screening can be performed as young as 2 years of age, and can be done at any age thereafter when the high-risk medical condition is diagnosed. Details for screening and management of high-risk medical conditions in pediatrics can be found in the American Heart Association guideline [17]. 

### 7.2. Universal Screening

However, selective screening alone has been shown to be inadequate in capturing all cases of severe dyslipidemia, particularly HeFH. The CARDIAC study of 10-year-olds in West Virginia demonstrated that using selective screening as the sole strategy for detecting severe pediatric dyslipidemia would have missed 37% of children who met criteria for pharmacotherapy [44]. Relying on family history in pediatrics can be particularly challenging as they are time consuming to acquire, and can be inaccurate, incomplete, or unavailable. Additionally, due to widespread use of statins in adults, reliance on a family history of premature cardiac events becomes less appropriate as premature events in children’s relatives are prevented. As such, universal screening of children is the most effective method to identify children with HeFH and other severe dyslipidemias [44,45].

Universal screening is recommended by the National Heart, Lung, and Blood Institute and endorsed by the American Academy of Pediatrics. The United States Preventive Services Task Force reviewed universal pediatric cholesterol screening in 2016 with an “I” recommendation, indicating the evidence was insufficient to recommend for or against screening [46]. Screening in children should be performed between 9–11 years old, before the onset of puberty. It may be useful to note that some racial backgrounds may have earlier onset puberty (such as African American females) and as such timing of screening should reflect this [47]. Screening can be performed with either a non-fasting or a fasting lipid panel, with the understanding that non-fasting lipid panels yielding extremely elevated results should be repeated as a fasting lab.

However, studies on current screening practices demonstrate that implementation of this recommendation has been slow, and most children are still not being screened for these disorders [48,49]. Furthermore, screening rates are potentially different among different types of pediatric clinicians, due to conflicting recommendations from guidelines [50]. 

### 7.3. Cascade Screening

Cascade screening is a method in which the family members of a patient diagnosed with a medical condition are subsequently screened for the disorder [51]. This is particularly ideal for autosomal dominant disorders such as HeFH, where relatives have a high probability of having the disorder [18]. This method has also been found to be cost-effective for HeFH detection when compared to the costs incurred from treating cardiovascular disease over time [52].

Cascade screening is traditionally done by screening children and other relatives of the index case. In pediatrics, this is often done through “reverse” cascade screening; diagnosing parents and other older relatives after lipid abnormalities are diagnosed in children. A cascade screening mentality is essential when diagnosing any patient with HeFH, and it is crucial to recognize that affected parents should have their children screened. As such, all adult patients identified with these disorders should be informed of the risk to first-degree relatives so that their family members, including their children, can be screened.

## 8. Conclusions

Although there has been increasing awareness of the need for screening and treatment of pediatric dyslipidemias, education and action from healthcare teams is still lagging, resulting in potential gaps in care. Lack of awareness of screening and treatment guidelines for pediatric dyslipidemia has even been seen among pediatric cardiologists [53], and is particularly true when considering HeFH [54]. As such, it remains critical that adults with HeFH or elevated cholesterol levels have their children screened. 

In this report are tools for all providers to utilize in their practice to help attain these goals. Universal pediatric lipid screening as well as screening family members of those diagnosed with severe dyslipidemia can help to further identify at-risk populations. Lifestyle modifications such as increasing intake of fruits and vegetables and elimination of sugar-sweetened beverages in children are good first steps in treatment. Finally, statins, the first-line pharmacotherapy for severe elevations in LDL-C, are rarely needed but are both safe and effective in lowering LDL-C in pediatric populations and thus lowering these children’s risk of future cardiac events. Responsibility to ensure the health of children and to decrease future morbidity and mortality from ASCVD lies with all healthcare providers, not solely those with a focus on pediatrics.

## Figures and Tables

**Table 1 jcm-11-06479-t001:** Reference Ranges for Pediatric Lipid and Lipoprotein Concentrations.

	Acceptable	Borderline	High
Total Cholesterol	<170	170–199	≥200
LDL-C	<110	110–129	≥130
HDL-C	>45	40–45	<40
Non-HDL-C	<120	120–144	≥145
Triglycerides			
0–9 years old	<75	75–99	≥100
10–19 years old	<90	90–129	≥130
Apolipoprotein B	<90	90–109	≥110
Apolipoprotein A-I	>120	115–120	<115

All values are in mg/dL.

**Table 3 jcm-11-06479-t003:** Lifestyle modifications for Pediatric Dyslipidemias [14,15,16]. Adapted from the following: 1. Williams, L.A.; Wilson, D.P. Nutritional Management of Pediatric Dyslipidemia. In Endotext; Feingold, K.R., Anawalt, B., Boyce, A., et al., Eds.; MDText.com, Inc.: 2000. Available online: http://www.ncbi.nlm.nih.gov/books/NBK395582/ (accessed on 30 June 2022). Piercy, K.L.; Troiano, R.P.; Ballard, R.M.; Carlson SA, Fulton JE, Galuska DA, George SM, Olson RD. The Physical Activity Guidelines for Americans. JAMA 2018, 320, 2020–2028. 3. U.S. Department of Agriculture; U.S. Department of Health and Human Services. Dietary Guidelines for Americans, 2020–2025, 9th ed.; 2020. Available online: DietaryGuidelines.gov (accessed on 30 June 2022).

Activity	
Increase physical activity	Recommend 60 min daily of moderate to vigorous physical activity which increases the heart rate, such as running, walking, dancing, biking, swimming, or sports like soccer or tennis.
Include muscle strengthening	Recommend 3 days per week, which can be integrated with the 60 min of daily activity above. This could include activities such as climbing on playground equipment, jumping rope, gymnastics, or skiing or snowboarding.
Diet	
Emphasize nutritionally dense foods	Encourage diets rich in a variety of fruits and vegetables, whole grains, proteins such as lean meat, seafood, and eggs, legumes, unsalted nuts and seeds, as well as dairy products including fat-free or low-fat options, yogurt, and cheese in appropriate portion sizes
Decrease saturated fat and trans fat intake	Saturated fats are included in red or fatty meats (such as sausage or bacon), high- fat dairy products, butter and other cooking fats. Trans fats are often found in processed foods and snacks including such as baked or fried goods.
Minimize sugar-sweetened beverages	Common sugar-sweetened beverages include soda, sports drinks, and coffee or tea drinks with added sugars. Consuming excess amounts of otherwise healthy beverages (such as fruit juice or chocolate milk) can be unwitting sources of sugar in the diet as well.
Increase beverages without added sugars	Encourage beverages such as water, fat-free or low-fat plain milk, or lactose free or fortified soy milk alternatives.
Eat the whole fruit	Try to eat fruits in whole forms when possible. While 100% fruit juice can be part of a healthy diet, it is lower in fiber than its whole fruit counterpart, and can be very calorie dense. Stick to the serving sizes to avoid excess sugar intake!
Behaviors	
Smoking	Counsel children and parents about smoking cessation and encourage against initiating smoking.

**Table 4 jcm-11-06479-t004:** Dosing and Expected Effect of Statins Currently Approved for Use in Children and Adolescents.

	Atorvastatin	Fluvastatin	Lovastatin	Pravastatin	Simvastatin	Rosuvastatin	Pitavastatin
Ages approved by FDA	≥10 years	≥10 years	≥10 years	≥8 years	≥10 years	≥8 years	≥8 years
Approved Pediatric Doses	5 mg, 10 mg, 20 mg, 40 mg	20 mg, 40 mg, 80 mg	10 mg, 20 mg, 40 mg, 80 mg	10 mg, 20 mg, 40 mg	5 mg, 10 mg, 20 mg, 40 mg	5 mg, 10 mg, 20 mg	1 mg, 2 mg, 4 mg
Expected LDL reduction at maximum pediatric dose (%) [23]	≥50%	30–49%	30–49%	30–49%	30–49%	≥50%	38%
Supplied as	10 mg, 20 mg, 40 mg, 80 mg tablets	20 mg and 40 mg capsules; 80 mg XR tablet	10 mg, 20 mg, 40 mg tablets; 20 mg, 40 mg, 60 mg XR tablets	10 mg, 20 mg, 40 mg, 80 mg tablets	5 mg, 10 mg, 20 mg, 40 mg, 80 mg tablets; Suspension: 20 mg/5 mL, 40 mg/5 mL	5 mg, 10 mg, 20 mg, 40 mg tablets; 5 mg, 10 mg, 20 mg, 40 mgsprinkle capsule	1 mg, 2 mg, 4 mg tablets
Notes		If LDL-C reduction ≥ 50% is needed, select a higher intensity statin (atorvastatin or rosuvastatin)	If LDL-C reduction ≥ 50% is needed, select a higher intensity statin (atorvastatin or rosuvastatin)	If LDL-C reduction ≥ 50% is needed, select a higher intensity statin (atorvastatin or rosuvastatin)	Simvastatin 80 mg should not be used due to myopathy risk. If LDL-C reduction goal cannot be achieved with simvastatin 40 mg, switch to higher intensity statin (atorvastatin or rosuvastatin)

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
