# Peer review of "Screening and Management of Dyslipidemia in Children and Adolescents"

_jcm, 2022, doi:10.3390/jcm11216479_

Round 1
Reviewer 1 Report
This is a very well written manuscript on an extremely important and current topic – screening of pediatric dyslipidemia. The authors presented comprehensively the characteristics of these disorders, screening and therapeutic approach.
I have only one suggestion, which concerns the part about atherogenic dyslipidemia. To make it completely clear, the presence of small, dense LDL particles should be added.
Including a bit of space between tables and the next paragraph would be helpful for orientation.
Author Response
Below are the comments from Reviewer 1 with point-by-point responses:
This is a very well written manuscript on an extremely important and current topic – screening of pediatric dyslipidemia. The authors presented comprehensively the characteristics of these disorders, screening and therapeutic approach. The authors appreciate the reviewer’s thoughtful review of our manuscript.
I have only one suggestion, which concerns the part about atherogenic dyslipidemia. To make it completely clear, the presence of small, dense LDL particles should be added. The authors added a sentence about small dense LDL to the Diagnosis paragraph in the Atherogenic Dyslipidemia section as requested.
Including a bit of space between tables and the next paragraph would be helpful for orientation. Spacing was added as requested.
Reviewer 2 Report
Very well-written and thorough review of pediatric dyslipidemias. One suggestion would be to include clear diagnostic criteria (e.g., actual lab cut-offs) for the various dyslipidemias, such as elevated Lp[a].
Author Response
Below are the comments from Reviewer 2 with point-by-point responses:
Very well-written and thorough review of pediatric dyslipidemias. One suggestion would be to include clear diagnostic criteria (e.g., actual lab cut-offs) for the various dyslipidemias, such as elevated Lp[a]. The authors appreciate the reviewer’s careful review of our manuscript. The authors have included a table of normal pediatric lipid values as a new Table 1, and added a statement to the section on Lipoprotein (a) to explain that there are no differences between pediatric and adult ranges for Lp(a).
Reviewer 3 Report
In this review article, the authors tried to briefly summarize diagnosis and treatment of dyslipidemia in children and adolescents. I agree with the authors that it is important for all physicians to aware the importance of identification of this situation in childhood. However, there are concerns listed below.
I think the authors should focus on some specific situations of dyslipidemia, because only few data and discussions are currently presented for each situation, including familial hypercholesterolemia (FH). For example, we need to consider many aspects when we want to diagnose and treat the patients with pediatric FH, such as difficulty of collecting family history information, lack pf physical findings, less awareness about the treatment, strategies for screening.
In addition, it is important to account for the differences with the situation of adults. What and how different with the adults?
There are so many issues need to be presented and discussed in obesity in childhood.
Author Response
Below are the comments from Reviewer 3 with point-by-point responses:
In this review article, the authors tried to briefly summarize diagnosis and treatment of dyslipidemia in children and adolescents. I agree with the authors that it is important for all physicians to aware the importance of identification of this situation in childhood. However, there are concerns listed below. The authors appreciate the reviewer’s review of the manuscript.
I think the authors should focus on some specific situations of dyslipidemia, because only few data and discussions are currently presented for each situation, including familial hypercholesterolemia (FH). For example, we need to consider many aspects when we want to diagnose and treat the patients with pediatric FH, such as difficulty of collecting family history information, lack pf physical findings, less awareness about the treatment, strategies for screening. The authors included discussions of several dyslipidemia patterns that are commonly encountered in pediatrics, as well as discussions of a few rare lipid disorders that may present to adult and pediatric lipid experts. Content regarding pediatric FH-related challenges with family history and lack of physical findings are already discussed in the Diagnosis section of HeFH, and the section Screening for Pediatric Dyslipidemias discusses selective, universal, and cascade strategies for screening.
In addition, it is important to account for the differences with the situation of adults. What and how different with the adults? The authors feel a point-by-point comparison between pediatric and adult aspects of dyslipidemia is beyond the scope of this manuscript, and expect that the readership of this journal likely already has familiarity with adult aspects of lipid management..
There are so many issues need to be presented and discussed in obesity in childhood. The authors agree that the pediatric obesity epidemic is of extreme importance. Given the space that would be required to appropriately discuss such a complex topic, the authors decided to focus on dyslipidemia diagnosis and management in pediatrics.
Round 2
Reviewer 3 Report
no additional comments.
Author Response
The authors appreciate the reviewer's thoughtful evaluation of our work.